# The Rare Phenomenon of Consecutive Ejaculations in Male Rats

**Joanna M. Mainwaring [1], Angela C. B. Garcia [1], Elaine M. Hull [2]** and **Erik Wibowo [1,***

[1] Department of Anatomy, University of Otago, Dunedin 9016, New Zealand; maijo090@student.otago.ac.nz (J.M.M.); garan337@student.otago.ac.nz (A.C.B.G.)

[2] Department of Psychology, Florida State University, Tallahassee, FL 32306, USA; hull@psy.fsu.edu

* Correspondence: erik.wibowo@otago.ac.nz

**Abstract:** Mounting, intromission and ejaculation are commonly reported sexual behaviours in male rats. In a mating session, they can have several copulatory series with post-ejaculatory intervals in between ejaculations before they reach sexual satiety. Here, we describe a phenomenon where male rats displayed consecutive ejaculations (CE) with a short inter-ejaculatory interval (IEI). Male rats were daily mated with a sexually receptive female rat. Two out of 15 rats displayed CE in one of their mating tests. The first rat had CE at 9.9 and 10.1 min (IEI = 16.3 s) after the start of the test. The second rat showed CE at 28.1 and 28.5 min (IEI = 18.7 s) after the test onset. During the IEI, the rats did not show any mounting or intromission.

**Keywords:** multiple ejaculations; male rodents; male sexual behaviour; multiple orgasms; consummatory behaviour; refractory period





## 1. Introduction

Male sexual behaviours in rodents are characterised by three distinct behaviours: mounting, intromission, and ejaculation [1,2]. During an ejaculation, there is a vaginal penetration (the deep forward pelvic thrust), and the male rat freezes on the female for one to three seconds [1]. While the actual semen expulsion is not usually visible, a plug can occasionally be found in the vagina or surrounding area because rat semen coagulates quickly to form a plug. Typically, a male rat could reach an ejaculation after a series of mounts and intromissions. Following ejaculation, the rat enters a refractory period, during which he does not respond to sexual stimuli for several minutes [3] before he resumes another series of mounts and intromissions until the next ejaculation. Male rats can have multiple copulatory series for about 150 min [4], ranging from 5 to 12 copulatory series [5]. After this, the rats will reach sexual satiety or sexual exhaustion, and remain sexually inactive for a prolonged period of time.

Recently, we conducted a study on the impact of chronic sleep deprivation (CSD) on male sexual behaviours in rodents [6]. The study involved sexually experienced male rats, which were subjected to CSD, imposed by keeping them awake for the last four hours of the light phase. Control rats were left undisturbed in their home cage at the same time of day. In that study, two of the rats (one from each group) showed two consecutive ejaculations (CE), separated by <20 s. In reviewing the literature for such behaviour, some studies have reported that rats are capable of having multiple ejaculations in a single mating session, and each pair of ejaculations is separated by a post-ejaculatory interval (PEI), as well as a series of mounts and intromissions [5,7]. However, we did not find any report stating that rats can have CE, with a short inter-ejaculatory interval (IEI), and no mounts or intromissions before the second ejaculation.

As in rodents, humans can also display ejaculation, i.e., expulsion of semen following penile stimulation (be it during solo masturbation or penetrative sex). However, humans

can also experience an orgasm, i.e., an intense, pleasurable response to genital or non-genital stimulation [8]. In humans, ejaculation and orgasm may be perceived as a single event, even though they are not the same biological process [9]. Furthermore, there is evidence that some men can have an ejaculation without having an orgasm [10,11], and some men can have an orgasm without an ejaculation [12–15]. There are also case reports on men who can have multiple ejaculations within a short period, but they required at least a few minutes of sexual stimulation between ejaculations [16,17]. However, to date, we are aware of no published report of men who can have ejaculations with no sexual stimulation (e.g., penile stimulation) before the second ejaculation. Despite this, one of us (EW) has received several anecdotal claims from men who reported having minimal or no refractory periods and are able to have multiple ejaculations with and without orgasms within a narrow time frame. This raises the question of whether the CE behaviour that we observed in our rats could be used as a model for multiple ejaculations and/or multiple orgasms in men.

The CE behaviour in rodents may bear a resemblance to men with minimal or short refractory periods, who can have multiple ejaculations. However, whether the CE behaviour can mimic multiple orgasms in men is difficult to answer, because we cannot assess orgasm in rats. Pfaus et al. [18] recently described how rats can have orgasm-like responses. For example, during ejaculation, male rats have pelvic floor contractions [19], as are also observed in humans [20]. Whether or not a rat experiences orgasm as a human does, in the presence of a receptive partner, a male rat will approach her in a way to maximise the 'reward' associated with ejaculation.

The purpose of this article is to describe a rare phenomenon where rats displayed two ejaculations consecutively with a short IEI, and no mounts or intromissions, between them. This behaviour was observed in one strain of inbred laboratory animals. To date, there is no documentation of whether other strains or non-laboratory rats can also show such behaviour.

## 2. Methods

### 2.1. Animals

All rats in this study were adult male Long Evans rats averaging $91.4 \pm 17.3$ days old on the first day of the experiment (i.e., first day of two weeks of daily mating tests). They were housed individually under a 14:10 light:dark cycle at $21 \pm 1$ °C ambient temperature, and had access to food and water ad libitum. The University of Otago Animal Ethics Committee approved this study protocol (AUP-19-135). The male rats had at least four encounters (30 min each, on separate days) with a receptive female before the first day of the experiment. Sexual experience was observed at 4 to 6 h into the dark phase of their lighting cycle (30 min each), under a sodium light. All tests were recorded on a digital camera. All rats were given up to six separate encounters to demonstrate ejaculation twice with one female rat in one session. In this study, only one rat of the 16 was excluded because he ejaculated fewer than twice in the last sexual experience session, meeting our criteria for hyposexual.

The female rats used in the mating tests were ovariectomised. Prior to surgery, they received an analgesic (5 mg/kg, carprofen) subcutaneously and a local anaesthetic (2 mg/kg bupivacaine) at the incision site. The surgery was then performed through an incision to the lateral walls of the abdomen under isoflurane anaesthesia (4% induction, 2% maintenance; 1 L/min).

The ovaries were removed from the abdomen using forceps, before the oviduct and associated blood vessels were clamped with haemostats. These were tied off with absorbable sutures and cut distal to the sutures. Absorbable sutures were used again to close the skin and abdominal muscle. Carprofen was delivered at the same dose as it was initially for two more days after surgery. The females did not undergo any behavioural testing for at least seven days following surgery, and they recovered in the animal care facility. Before a

mating test, they received subcutaneous injections of estradiol benzoate (20 μg) 48 h before testing, followed by progesterone (500 μg) 4 h before testing, to induce sexual receptivity.

Male rats were allowed to have a few days of no mating before the main experiment (involving daily mating tests for 14 days) started. For the main experiment, the rats were randomly grouped into either those that would receive CSD and those that would not (control). Rats assigned to the CSD group were deprived of sleep for the last 4 h of the light period. This was carried out for seven consecutive days, followed by a seven-day recovery period. The rats not allocated to the CSD group remained in their home cage and experienced no sleep disruption for the 14 days. CSD was achieved by disrupting the rat whenever it showed signs of getting ready to sleep (lying still, for example, or curling up). Such methods included shaking the bedding, tapping the cage, or introducing new objects (plastic toys). No physical contact with the rat was made. Previously, when the same protocol had been used, rats remained awake for 98% of the desired period [21]. Following the conclusion of all behavioural tests, all rats were euthanised.

*2.2. Data Analyses*

The recordings of the rat behaviour taken throughout the experiment were scored using BORIS software [22], specifically mounting, intromission, and ejaculation. These were as defined in previous studies [1,23,24]. Data calculated from this included the frequencies of each behaviour (total number of specific behaviours within the 30-min test) and the latencies (time from the beginning of the test to the first instance of the behaviour). The time between ejaculation and subsequent intromission was analysed as the post-ejaculatory interval (PEI). In this article, descriptive statistics were used to describe the sexual behaviour of rats that showed CE, and those that did not (non-CE). Each of the ejaculations during CE was indistinguishable from typical ejaculation behaviour by male rodents, as described previously [1]. However, we cannot confirm whether there was an ejaculate released during each ejaculation. Due to the limited sample size, we are unable to conduct quantitative statistical tests.

## 3. Results

Among the 15 rats in the main experiment, two rats displayed CE. Each of these behaviours appeared to be what is normally considered an ejaculation in a standard mating test; i.e., a deep pelvic thrust, with the male remaining on the female for one to three seconds. For each rat, the CE happened in only one out of 18 mating tests. The first rat was from the CSD group. That rat showed CE in the second daily test (at 98 days old), whereas the second one was from the control group, and he had CE in the fifth daily test (at 71 days old). Their sexual behaviour parameters on the day they showed CE are shown in Table 1. The average sexual parameters of non-CE rats on the same day are also presented for comparison.

The first rat had ejaculations at 9.9 and 10.1 min after the start of the test. The time gap between these ejaculations was 16.3 s, and during this time, he did not show any other sexual behaviour, but he groomed his genitals briefly. He also had another ejaculation at 25.8 min after the test onset (i.e., 15.2 min after the last ejaculation).

The second rat had a single ejaculation at 9.9 min, followed by another ejaculation at both 28.1 and 28.5 min after the start of the test. The time gap between the latter two ejaculations was 18.7 s, and during this time, he did not show any other sexual behaviour or genital grooming.

**Table 1.** Sexual behaviour of male rats on the day when they displayed CE. Parameters of non-CE rats on the same day (mean ± standard deviation) are also presented next to each CE rat's parameters.

| Sexual Behaviour Parameters | Test on Day 2 | | Test on Day 5 | |
|---|---|---|---|---|
| | **Rat 1** | **Non-CE Rats** | **Rat 2** | **Non-CE Rats** |
| *First copulatory series* | | | | |
| Mounting frequency | 30 | 13.9 ± 9.2 | 12 | 14.6 ± 12.9 |
| Intromission frequency | 8 | 14.5 ± 3.4 | 19 | 15.6 ± 5.7 |
| Ejaculation frequency | 2 | 1 | 1 | 1 |
| Mounting latency (seconds) | 7.5 | 17.8 ± 24.2 | 7.5 | 48.8 ± 86.1 |
| Intromission latency (seconds) | 129.2 | 57.3 ± 148.1 | 10.5 | 84.7 ± 158.6 |
| Ejaculation latency (seconds) | 462.3 | 439.2 ± 165.7 | 584.8 | 487.3 ± 281.8 |
| PEI (seconds) | 145.0 | 475.0 ± 184.3 | 856.7 | 710.2 ± 188.6 |
| *Second copulatory series* | | | | |
| Mounting frequency | 26 | 9.0 ± 4.3 | 3 | 5.8 ± 4.3 |
| Intromission frequency | 1 | 8.7 ± 4.2 | 9 | 7.0 ± 1.6 |
| Ejaculation frequency | 1 | 1 | 2 | 0.9 ± 0.3 |
| Mounting latency (seconds) | 36.0 | 466.2 ± 180.1 | 843.5 | 674.5 ± 191.8 |
| Intromission latency (seconds) | 145.0 | 475.6 ± 186.1 | 856.7 | 710.2 ± 188.6 |
| Ejaculation latency (seconds) | 809.0 | 264.5 ± 120.5 | 236.3 | 202.8 ± 89.3 |

## 4. Discussion

For the first time, we report CE in male rats. Currently, it remains unknown what the neurobiological basis is for them to show such behaviour, such as hormonal or neurotransmitter factors that can affect the ability to have CE. In addition, it remains unclear to what extent penile grooming post-ejaculation plays a role in stimulating the second ejaculation during CE. Sensory input from the genitals during genital grooming may be conveyed to the spinal ejaculation generator in the lumbar spinal cord [25]. However, one of the CE rats groomed his genitals in between ejaculations, but the other one did not.

We are aware of no published data on similar behaviour in humans either. As noted above, one of us (EW) received several anecdotal claims from men with no or minimal refractory periods, who reported CE. However, there are published data that some men reported multiple orgasms with ejaculations with intervals of several minutes between them [9]. Undoubtedly, the volume of the ejaculate decreases with subsequent ejaculations [16]. Another observation in a man with such an ability showed that the person did not have orgasm/ejaculation-induced prolactin release [17]. It remains to be determined whether the dampening of prolactin release after ejaculations may play a role in the CE in rats. Another potential mechanism for the CE we observed may involve a change in the serotonergic system. Past studies indicate that the administration of a 5-HT1A receptor agonist reduces the number of intromissions before ejaculation and shortens the ejaculation latency in male rats [26,27]. Considering that the animals in our study were not treated with any serotonergic agents, there is a possibility that they may have individual variation in their serotonergic system.

Future study in this area will be challenging, given the rarity of the phenomenon, and we cannot predict when they would show such a behaviour. One potential future study would be to explore hormonal (e.g., on prolactin) and neurobiological (e.g., on the serotonergic system) differences between rats who are and are not capable of having CE. It would be interesting to investigate whether prolactin receptor knock-out rodents display CE at an elevated frequency compared to wild-type rodents. Another possibility is to test whether some psychostimulants can increase the frequency of displaying CE, because some men have reported having multiple orgasms while having sex under the influence of psychostimulants, although it is unclear if they also had ejaculations [9].

## 5. Conclusions

In conclusion, in rare cases, male rats are capable of displaying CE, with a short (<20 s) interval between ejaculations. It is now unclear how often such behaviour is observed in rats. Potentially, the frequency may vary based on the rat strains. The CE behaviour should be considered for future studies that assess male sexual behaviour in rodents. In addition, future studies could aim to determine the biological mechanism underlying this phenomenon.

**Author Contributions:** Conceptualization E.W.; formal analysis E.W., A.C.B.G., J.M.M., E.M.H.; funding acquisition E.W.; investigation E.W., A.C.B.G., J.M.M., E.M.H.; methodology E.W., A.C.B.G., J.M.M., E.M.H.; project administration E.W.; resources E.W.; supervision; roles/writing—original draft E.W.; writing—review and editing E.W., A.C.B.G., J.M.M., E.M.H. All authors have read and agreed to the published version of the manuscript.

**Funding:** This study was supported by a fund from the Department of Anatomy at the University of Otago.

**Institutional Review Board Statement:** The University of Otago Animal Ethics Committee approved this study protocol (AUP-19-135).

**Data Availability Statement:** Data are not deposited in a repository. The data presented in this study are available on request from the corresponding author.

**Acknowledgments:** We thank Matt Newdick for taking care of our rats.

**Conflicts of Interest:** The authors declare no conflict of interest.

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
