# Peer review of "The Rare Phenomenon of Consecutive Ejaculations in Male Rats"

_sexes, doi:10.3390/sexes2020016_

Round 1
Reviewer 1 Report
Dear authors,
I read the revised manuscript and found the authors' replies to be satisfactory. However, I still think the video recordings to be of value to my final decision. I do not think the authors' analysis is flawed, instead, I think the opinion of a researcher not involved directly with the study to be of importance and help to strengthen the occurrence of 'consecutive ejaculations'. Sure I do recognize the experience of Dra Elaine Hull who must be regarded as one of the founders of neurochemistry/neuroanatomy of ejaculation behavior. Actually, I would like to express my gratitude to Dra Hull as her published studies were between the ones that shaped (and still shaping) my interest in male sexual function and the projects in progress in my lab. Additionally, Dra Hull's work on male rodent sexual behavior (Hull & Dominguez Horm Beh 52:45, 2007) figures out as a required material to students in my lab. After analysis of copulatory behavior of hundreds of male rats (I still in activity in behavioral experiments) I have never faced a consecutive ejaculation behavior but I must confess maybe I was not able to recognize it. That's why I insist on getting access to the video recordings as they can be of value to improve my own analysis (and of my current and future students) and the analysis of future readers of the putative paper.
Overall I am not comfortable in accepting the description of the behavior without analyzing the video recordings.
My best wishes,
anonymous reviewer 1.
Author Response
We have now provided the videos via the Editor. Please let us know if Reviewer 1 has additional questions.
Reviewer 2 Report
The changes made to the text and their responses to the reviewer's comments are appropriate. I judge that this manuscript is now worth being published.
Author Response
Thank you for the positive feedback.
Round 2
Reviewer 1 Report
Dear authors,
I have evaluated the provided video recordings of male rat copulatory behavior and found it to be consistent with the proposed behavioral consecutive ejaculations in rats. As requested, all the video files provided were deleted.
Kind regards,
anonymous reviewer 1
This manuscript is a resubmission of an earlier submission. The following is a list of the peer review reports and author responses from that submission.
Round 1
Reviewer 1 Report
In the manuscript entitled “A rare phenomenon of consecutive ejaculations in male rats” by Mainwaring and colleagues the authors suggest the observation of consecutive ejaculations (CE) with minimal sexual stimulation in the rat. The study is descriptive in nature requiring no deep explanation on possible underlying physiological mechanisms (albeit the authors attempt to at least suggest possibilities in the discussion section).
Overall, on the basis of the presented data it is not possible to ascertain about the putative CE behavior. I think a better description of the proposed behavior is required addressing at least the following two major (and basic) points: 1) how did the rats behave during the “consecutive ejaculation” behavior? 2) what is the evidence that the proposed behavior in fact represents an ejaculation?
I think the video recordings of the putative “successive ejaculations” are of outstanding importance to assist the review process as no irrefutable evidence for the CE behavior is found throughout the text. Bellow I highlight issues to be addressed by the authors.
Major issues:
>>Page 1, Abstract section: “The rats’ CE behaviour may potentially be used as a model for men who can have multiple ejaculations with a short IEI”
The sentence is overstated as neither the rat’s CE behavior nor the putative human counterpart are clearly defined. Please remove.
>> A clear description of the sexual experience Schedule is required. It it stated that “The male rats had at least 4 encounters (30 minutes each) with a receptive female on the first Day of the experiment...”
Were rats exposed up to 4-6 times to the same/different female until they copulate to 2 ejaculations?
Was this method of multiple encounters followed in the next daily mating sessions?
>>Page 2: “In this study, only 1 rat of the 16 was excluded because he ejaculated fewer than twice in a mating session, meeting our criteria for hyposexual.”
Was this exclusion criteria applied to rats in the first mating session (1st day) only?
Data analysis section
>>Page 2,: “The recordings of the rat behaviour taken throughout the experiment were scored using BORIS software [20], specifically mounting, intromission, and ejaculation. These were as defined in previous studies [1,21,22].”
As the authors know, immediately after the ejaculation in copulatory studies the rat engages in genital grooming which is followed by non-genital self-grooming in areas such as the nose, face, head, ears, the ventrolateral region of the trunk and tail and then lay down in inactivity for most of the refractory period. All the post-ejaculatory grooming period takes about a minute or so, therefore, the authors’ scored sucessive ejaculation must be occuring during the genital grooming period. I would like to know how was the “consecutive” ejaculation detected, i.e. how did the rats behave? what is the evidence that it does indeed reflect an ejaculation? Actually, I would like to request the authors to submit the video recordings of consecutive ejaculations to assist the review process; the same video recordings must also be made available as a supplementary file in the case the manuscript is accepted.
>>Page 3: While is stated in the Material and Methods section that rats averaged 91 days at the start of the first mating test, it is the Rat 2 would be at 69 days-old in the first mating test. As male rats are considered adult at 90 days-old it argues to a more careful method in future studies on rat ejaculation behavior.
>>Page 3, Table 1: Data of Rat 1 is confusing. The authors state the Rat 1 required only 1 intromission to attain the second ejaculation, is that correct? While is known the number of intromissions required to attain the ejaculations in successive ejaculatory series is significantly lower, on average half the number of the first ejaculatory series as shown by the authors own data, a single intromission would be highly uncommon (unless pharmacologically facilitated as shown to 8-OH-DPAT). In addition the mount and intromission latencies in the second ejaculatory series are very short as well (36s and 146s). I question whether the data presented were correctly typed as an ejaculation latency of 954s to the second ejaculation is reported; as the ejaculation latency is defined by the interval between the first intromission until the ejaculation, an intromission frequency of 1 would represent an ejaculation latency of zero or the time spent in the ejaculating intromission.
>>Page 3: I would like to suggest presenting the whole data to Rat 1 and 2 across the different mating tests in parallel to the average of Non-CE Rats in the form of graphs to allow a full appreciation of Rat 1 and 2 copulatory behavior.
>>Discussion section: Considering the penile grooming behavior characteristic of rats immediately after the ejaculation is able to carry information to the spinal generator of ejaculation, the authors must include a discussion about the possibility of such kind of penile stimulation as a contributor to the induction of “consecutive ejaculation” behavior.
Author Response
We thank the Editor for the opportunity to revise our paper, and the Reviewers for their feedback. We have now revised our paper, and hope that the current version is acceptable for publication.
Reviewer 1
In the manuscript entitled “A rare phenomenon of consecutive ejaculations in male rats” by Mainwaring and colleagues the authors suggest the observation of consecutive ejaculations (CE) with minimal sexual stimulation in the rat. The study is descriptive in nature requiring no deep explanation on possible underlying physiological mechanisms (albeit the authors attempt to at least suggest possibilities in the discussion section).
Overall, on the basis of the presented data it is not possible to ascertain about the putative CE behavior. I think a better description of the proposed behavior is required addressing at least the following two major (and basic) points:
1) how did the rats behave during the “consecutive ejaculation” behavior?
The rats showed what is normally considered ejaculation in a standard mating test encounter. There was a deep pelvic thrust, and the male remained on the female for 1-3 seconds. We have added this detail in the Results.
2) what is the evidence that the proposed behavior in fact represents an ejaculation?
The behaviour was indistinguishable from typical ejaculation behaviour by male rodents, as described in: Agmo A. Male rat sexual behavior. Brain Res Protoc. 1997;1:203-9. Whether there was an ejaculate release, we cannot be certain. We have added this detail in the Results.
I think the video recordings of the putative “successive ejaculations” are of outstanding importance to assist the review process as no irrefutable evidence for the CE behavior is found throughout the text. Bellow I highlight issues to be addressed by the authors.
Major issues:
>>Page 1, Abstract section: “The rats’ CE behaviour may potentially be used as a model for men who can have multiple ejaculations with a short IEI”
The sentence is overstated as neither the rat’s CE behavior nor the putative human counterpart are clearly defined. Please remove.
Done.
>> A clear description of the sexual experience Schedule is required. It it stated that “The male rats had at least 4 encounters (30 minutes each) with a receptive female on the first Day of the experiment...”
Were rats exposed up to 4-6 times to the same/different female until they copulate to 2 ejaculations?
Each rat was allowed to have 2-3 ejaculations with one female rat in a single session. This detail is now added.
Was this method of multiple encounters followed in the next daily mating sessions?
Not immediately. They were allowed to have a few days of no mating before the daily mating started. We have added this detail in the Results.
>>Page 2: “In this study, only 1 rat of the 16 was excluded because he ejaculated fewer than twice in a mating session, meeting our criteria for hyposexual.”
Was this exclusion criteria applied to rats in the first mating session (1st day) only?
No. This exclusion criterion was applied to the last sexual experience session. We have updated the relevant sentence.
Data analysis section
>>Page 2,: “The recordings of the rat behaviour taken throughout the experiment were scored using BORIS software [20], specifically mounting, intromission, and ejaculation. These were as defined in previous studies [1,21,22].”
As the authors know, immediately after the ejaculation in copulatory studies the rat engages in genital grooming which is followed by non-genital self-grooming in areas such as the nose, face, head, ears, the ventrolateral region of the trunk and tail and then lay down in inactivity for most of the refractory period. All the post-ejaculatory grooming period takes about a minute or so, therefore, the authors’ scored sucessive ejaculation must be occuring during the genital grooming period. I would like to know how was the “consecutive” ejaculation detected, i.e. how did the rats behave? what is the evidence that it does indeed reflect an ejaculation?
As mentioned above, we cannot be certain that there was any ejaculate released. However, the behaviour resembles typical ejaculations where there were deep pelvic thrusts followed by 1-3 seconds of not moving while on the female. One of the rats licked his genitals during the brief (<20 seconds) inter-ejaculatory interval, while the other one did not. These details are now added in the paper.
Actually, I would like to request the authors to submit the video recordings of consecutive ejaculations to assist the review process; the same video recordings must also be made available as a supplementary file in the case the manuscript is accepted.
We have sought feedback from our ethic committee for this request. At this stage, we do not feel it necessary to attach the videos as a supplementary file in the approved publication. However, we can share the videos to Reviewer 1 if the Editor wishes. All authors have conducted multiple testings on rat sexual behaviour. Furthermore, one of the authors (EH) is an expert in the field, and has done research on male sexual behaviour in rodents, dating back to 1970s. She has confirmed that the behaviours indeed appear like typical ejaculations.
If the Editor would like us to send the file to Reviewer 1. We can send it to the Editor via a separate email. The file size is over 10MB each, so we will need to use a specialized secure transfer method, recommended by our institution. In addition, our local ethic committee wants an assurance from Reviewer 1 and the Editor that the videos are deleted once the reviewer views the video.
>>Page 3: While is stated in the Material and Methods section that rats averaged 91 days at the start
of the first mating test, it is the Rat 2 would be at 69 days-old in the first mating test. As male rats are considered adult at 90 days-old it argues to a more careful method in future studies on rat ejaculation behavior.
Thank you for this feedback. Rats typically have completed their puberty at around day 55, so all of our rats were young adults by the time they had their first sexual encounter.
>>Page 3, Table 1: Data of Rat 1 is confusing. The authors state the Rat 1 required only 1 intromission to attain the second ejaculation, is that correct? While is known the number of intromissions required to attain the ejaculations in successive ejaculatory series is significantly lower, on average half the number of the first ejaculatory series as shown by the authors own data, a single intromission would be highly uncommon (unless pharmacologically facilitated as shown to 8-OHDPAT).
This is correct. We have double checked the video recording to confirm that Rat 1 only had 1 intromission in the second copulatory series.
In addition the mount and intromission latencies in the second ejaculatory series are very short as well (36s and 146s).
These are correct. The rat started mounting again soon after the CE.
I question whether the data presented were correctly typed as an ejaculation latency of 954s to the second ejaculation is reported; as the ejaculation latency is defined by the interval between the first intromission until the ejaculation, an intromission frequency of 1 would represent an ejaculation latency of zero or the time spent in the ejaculating intromission.
Thank you for pointing this out. We have corrected the numbers.
>>Page 3: I would like to suggest presenting the whole data to Rat 1 and 2 across the different mating tests in parallel to the average of Non-CE Rats in the form of graphs to allow a full appreciation of Rat 1 and 2 copulatory behavior.
We thank the reviewer for this suggestion. However, we do not feel that presenting such data will bring in any new relevant information, because Rat 1 and Rat 2 did not show CE on other days.
>>Discussion section: Considering the penile grooming behavior characteristic of rats immediately after the ejaculation is able to carry information to the spinal generator of ejaculation, the authors must include a discussion about the possibility of such kind of penile stimulation as a contributor to the induction of “consecutive ejaculation” behavior.
We have added a section on penile grooming. We also would like to point out that only one of the rats showing CE groomed his penis during the inter-ejaculatory interval.
Reviewer 2 Report
The paper reports a phenomenon of consecutive ejaculations in male rats. The paper is well written with a clear description of introduction, methods, results, and discussion, however I have several questions that need to be addressed before this paper is considered for publication.
I found a couple of minor things to be modified.
Page 3 line 13 "quantitate" should be "quantitative"
Table 1 Indicate if the parameters are shown as mean and SD or else.
Page 1 line -1
What does “sexual activity” actually refer to? Does it refer to mounting and intromission? It seems that in the present paper it is assumed that mounting and intromission always precedes ejaculation of both rodents and men. Would there be any case for example that ejaculation happens following solely erotic imaginations without physical stimulus?
Page 2 lines 4 and 5
It is not clear to me what “could be used as a model for multiple ejaculations and/or multiple organisms” means. What is the significance of studying multiple ejaculations or organisms in men? Does it have any effect on fertility or sperm competition or any other phenotype? Do you assume that the mechanisms underlying multiple ejaculations and organisms in rodents and in men are common?
Can you state the definitions of orgasm and ejaculations for humans and rodents separatly?
The authors reported CE in male rats for the first time. How can we interpret this fact? Is it just because some researchers did encounter with such behavior, but just did not report it, or is it really a new behavior that nobody has reported before? If the latter is correct, why did these rats show such a strange behavior? Considering that CE were separated by a short time duration of <20 seconds, wouldn’t it possible that such behavior used to be counted as one ejaculation by other researchers?
Could you explain what is the merit of using BORIS software compared to direct observation by humans? If a person observes, would the CE be observed in the same way as by BORIS software? What is the algorithm of detecting ejaculation using BORIS software? Would it be possible for a person to observe ejaculation?
Author Response
We thank the Editor for the opportunity to revise our paper, and the Reviewers for their feedback. We have now revised our paper, and hope that the current version is acceptable for publication.
The paper reports a phenomenon of consecutive ejaculations in male rats. The paper is well written with a clear description of introduction, methods, results, and discussion, however I have several questions that need to be addressed before this paper is considered for publication. I found a couple of minor things to be modified.
Page 3 line 13 "quantitate" should be "quantitative"
This has now been corrected.
Table 1 Indicate if the parameters are shown as mean and SD or else.
This has now been added.
Page 1 line -1 What does “sexual activity” actually refer to? Does it refer to mounting and intromission? It seems that in the present paper it is assumed that mounting and intromission always precedes ejaculation of both rodents and men.
We have now replaced this with “sexual stimulation”. Humans do not exhibit mounting or intromission like rodents. But men may exhibit sexual stimulation (e.g., penile stimulation) in between ejaculations.
Would there be any case for example that ejaculation happens following solely erotic imaginations without physical stimulus?
We are aware of no published report on this. Suffice to say, there are anecdotal reports that some people can reach an orgasm without any physical stimulation. See: Herbenick D, Barnhart K, Beavers K, Fortenberry D. Orgasm Range and Variability in Humans: A Content Analysis. Int J Sex Health. 2018;30:195-209.
Page 2 lines 4 and 5, It is not clear to me what “could be used as a model for multiple ejaculations and/or multiple organisms” means. What is the significance of studying multiple ejaculations or organisms in men? Does it have any effect on fertility or sperm competition or any other phenotype? Do you assume that the mechanisms underlying multiple ejaculations and organisms in rodents and in men are common?
Studying such a phenomenon would help us better understand the variability in male sexual function. As noted in the paper, the PI (EW) has anecdotally received feedback from a few men claiming to be capable of having multiple ejaculations within a narrow time frame. The implication would be more of a social one. One man reported that his partner sometimes feels she cannot keep up with him, as he often attempts to have more than one ejaculations during a sexual activity session. These anecdotes invite more research on the neurobiological mechanism of multiple ejaculations. To date, the prevalence of men, who can have multiple ejaculations, is not known. Undoubtedly there are those who can have more than one ejaculation because of the Coolidge Effect, for example those engage in sexual activity with more than one partners.
Regarding the second question, we do not feel that there is any impact on fertility per se. The amount of ejaculate decreases in subsequent ejaculations: Whipple et al. J Sex Educ Ther. 1998;23:157-62.
As for the last question, we noted in the paper that we cannot study orgasm in rodents, but we can only assume that they may experience one given they experience similar physiological responses as orgasms in human like pelvic floor spastic contraction (Pfaus et al. Do rats have orgasms? Socioaffect Neurosci Psychol. 2016;6:31883). In contrast, ejaculation in both humans and rodents involve semen expulsion. Neurobiologically there may be some similarities and differences. For example, after an ejaculation there is neuronal activation in the brain of male rodents (particularly in preoptic area, bed nucleus or the stria terminalis and amygdala). But fMRI studies in human show inconsistent findings [cf. Holstege et al. J Neurosci. 2003;23:9185-93; Mallick et al. Indian J Physiol Pharmacol. 2007;51:81-5]. Furthermore, a spinal ejaculation generator has been found in rodents: Truitt and Coolen. Science. 2002;297:1566-9.] However, we are unaware of studies confirming such neurons in human. Understandably there are technical limitation in studying the neurobiology of ejaculation in human to compare with rodents.
Can you state the definitions of orgasm and ejaculations for humans and rodents separatly?
We have now added further description on ejaculation in human and rodent.
The authors reported CE in male rats for the first time. How can we interpret this fact? Is it just because some researchers did encounter with such behavior, but just did not report it, or is it really a new behavior that nobody has reported before? If the latter is correct, why did these rats show such a strange behavior?
We are not sure why this behaviour has not been reported previously. However, we do not feel that this is a new behaviour. One possibility is that researchers may not think such a behaviour is possible, and consider CE as intromission-ejaculation rather than ejaculation-ejaculation. On a separate topic, we have another ongoing study where we have screened 25 rats of the same strain from the same supplier, and two of them showed CE. We’d also would like to point out that the rats do not show CE in every test. Each of them only showed the behaviour in 1 out of 12-18 mating tests. Thus, past researchers may also not detect the behaviour if they only tested the rats a few times during the study.
Considering that CE were separated by a short time duration of <20 seconds, wouldn’t it possible that such behavior used to be counted as one ejaculation by other researchers?
We don’t think this is the case, because two behaviours within 20 seconds are quite distinguishable. One possibility is that the initial ejaculation may have been counted as an intromission.
Could you explain what is the merit of using BORIS software compared to direct observation by humans? If a person observes, would the CE be observed in the same way as by BORIS software? What is the algorithm of detecting ejaculation using BORIS software? Would it be possible for a person to observe ejaculation?
The BORIS Software is used for scoring purpose, but it does not involve an automated scoring. A person still needs to view the video recording of the mating tests and indicates the various sexual behaviour as a video is playing. The advantage of using the BORIS software is that we can get accurate information on time (i.e., latencies for the various behaviours). In person scoring would only provide an information on the behavioural frequencies. In addition, using the BORIS coupled to the video recording would allow researchers to re-watch the videos if they need to confirm any behaviours.